# Impact of the *Escherichia coli* Heat-Stable Enterotoxin b (STb) on Gut Health and Function

**DOI:** 10.3390/toxins12120760

**Published:** 2020-12-02

**Authors:** Shahnawaz Butt, Mazen Saleh, Jeffrey Gagnon

**Affiliations:** Department of Biology, Laurentian University, 935 Ramsey Lake Rd., Sudbury, ON P3E 2C6, Canada; sbutt1@laurentian.ca (S.B.); msaleh@laurentian.ca (M.S.)

**Keywords:** *Escherichia coli*, ETEC, enterotoxigenic, enterotoxins, heat-stable, gastrointestinal, internalization, tight junction, apoptosis, hormones

## Abstract

Enterotoxigenic *Escherichia coli* (ETEC) produces the heat-stable enterotoxin b (STb), which is responsible for secretory diarrhea in humans and animals. This toxin is secreted within the intestinal lumen of animals and humans following ETEC colonization, becoming active on enterocytes and altering fluid homeostasis. Several studies have outlined the nature of this toxin and its effects on gut health and the integrity of the intestinal epithelium. This review summarizes the mechanisms of how STb alters the gastrointestinal tract. These include the manipulation of mucosal tight junction protein integrity, the formation of enterocyte cellular pores and toxin internalization and the stimulation of programmed cell death. We conclude with insights into the potential link between STb intoxication and altered gut hormone regulation, and downstream physiology.

## 1. Introduction

Infectious diarrhea is one of the leading causes of death in children of developing countries [1]. Enterotoxigenic *Escherichia coli* (ETEC) is known to cause travelers’ diarrhea and diarrhea in farm animals [2]. The global burden of ETEC infection is extensive and remains a major public health issue. In 2006, the World Health Organization estimated annually 400–500 million diarrheal episodes in newborns and children and 400 million in individuals 15 years or older, all ETEC related [3]. ETEC can be characterized by its capability to produce and secrete several virulence factors, including colonization factors (adhesins) and enterotoxins [4]. Adhesins allow for the attachment of the bacteria to intestinal epithelial cells. These can be characterized as filamentous appendages on the surface of the bacteria called fimbriae, on which adhesins can be found [5]. The loss of these colonization factors may render the bacteria unable to colonize and cause infection [6]. Colonization of the intestinal tract allows for specific delivery of enterotoxins which disrupt normal intestinal fluid homeostasis [7]. ETEC enterotoxins are differentiated by their heat stability. They are classified into heat-labile (LT-I and LT-II) and heat-stable (STa and STb) enterotoxins [5]. This review will focus on the STb toxin specifically. For in depth reviews on STa and heat labile toxins please see [8] and [2].

The heat-stable enterotoxin b (STb) is one of several toxins secreted by ETEC responsible for inducing diarrhea [9]. STb is synthesized by ETEC as a pre-polypeptide consisting of 71 amino acids. The signal peptide is then cleaved within the periplasmic space, yielding a mature 48 amino acid peptide of 5.2 kDa [10]. This mature STb peptide is comprised of two α-helices affixed by two disulfide bridges (Figure 1). STb is heat-stable, possessing stable toxigenic properties up to 100 °C [2]. It is a fast acting toxin, inducing a diarrheal response of moderate extent. Within mouse intestinal loops, it was found that STb can trigger a fluid response within just 30 min, with fluid accumulation reaching its maximum at 3 h, ceasing after 16 h [11].

Upon ETEC colonization and enterotoxin secretion, STb-mediated intoxication begins by STb binding to its receptor, sulfatide. This is an acidic glycosphingolipid located on the surface of intestinal epithelial cells [14]. The internalization of STb by the cells results in the activation of a pertussis toxin-sensitive GTP-binding regulatory protein (Gαi3). This leads to an influx of extracellular calcium ions into the cell by means of a receptor-dependent ligand-gated calcium channel [15]. This increase in intracellular calcium activates calmodulin-dependent protein kinase II (CAMKII) through a calmodulin-calcium^2+^ complex. CAMKII activation leads to the opening of a calcium-activated chloride channel (CaCC), allowing for the secretion of chloride from the cell into the lumen [16]. Protein kinase C (PKC) and downstream cystic fibrosis transmembrane conductance regulator (CFTR) are also activated, resulting in the secretion of H_2_O, HCO_3_^+^ and various electrolytes from the cell. PKC also inhibits sodium uptake by acting upon an unidentified sodium channel deemed NHE3, increasing luminal sodium concentrations [14,15]. Increased intracellular calcium levels may also stimulate phospholipases A2 and C, releasing arachidonic acids from phospholipids present within the cell membrane. This step permits the formation of prostaglandin E_2_ (PGE_2_) and 5-hydroxytryptamine (5-HT). These secretagogues also mediate the transport of water and electrolytes form the cell into the lumen by an unknown mechanism (Figure 2) [17].

There are many gut functions which can become negatively altered by the STb toxin during an ETEC infection. This toxin has been observed to effect numerous animals, as well as humans. This review addresses the impacts of STb on overall gut function, outlining the various physiological changes that can occur, and the resulting alterations on overall gut health. Additionally, the effects of various gut hormones on intestinal function will be discussed, and how the regulation of these hormones may become altered during STb intoxication. This realm of bacterial toxin-induced gut hormone dysregulation has not yet been discussed in the literature. This is imperative, as gut hormone regulation is essential for overall gut health and bodily functions. 

## 2. Impact of ETEC STb on Host Gut Health and Function

### 2.1. STb Alters Barrier Function through Tight Junction Manipulation

Within the intestine, the luminal surface consists of a dynamic layer of epithelial cells. It acts as a barrier between luminal materials and the underlying neuronal and immune systems, also sustaining nutrient, fluid and ion transport [18,19]. This barrier is upheld by conjunctions between adjacent enterocytes. These are intercellular junctions known as adherens junctions (AJs) and tight junctions (TJs). TJs are multi-protein complexes that form a continuous loop-like ring around cells at the apical and lateral membrane domains. These proteins function as a selective paracellular barrier, facilitating the flow of solutes and ions through the lateral intercellular space. They prevent the entrance and transfer of intestinal antigens, microorganisms and their potentially harmful products [20]. TJs are comprised of four unique groups of transmembrane proteins: claudins, occludin, zonula occludens protein (ZOs) and junctional adhesion molecules (JAMs) [21]. Transmembrane claudins and occludin are supported by scaffolding ZO proteins. The ZO complex provides intracellular structural support for the claudin and occludin multiprotein complex at the cytoplasmic surface. (Figure 3a) [22].

Enteric pathogenic bacteria have crafted various methods in order to disrupt TJ complexes, either by cellular cytoskeleton alteration or by altering specific proteins within the TJ complex. The alteration of specific TJ proteins occurs either through degradation by bacterial proteases or through the biochemical phosphorylation/dephosphorylation pathways [18]. Transepithelial electrical resistance (TER) functions by establishing electrochemical gradients, and can be a measurement for the degree of tightness of these TJs. Paracellular flux is a measure for transport efficiency over time and can be a measurement for paracellular permeability [7,23]. With regards to the STb toxin, Ngendahayo and Dubreuil (2013) [7] demonstrated how purified STb triggered a significant reduction of TER alongside an increase in paracellular permeability in a human colonic cell line. Alterations in F-actin stress fibers were accompanied by this increase in paracellular permeability. Condensation and dissolution of these F-actin filaments were accompanied by a redistribution and/or fragmentation of specific TJ proteins occludin, claudin-1 and ZO-1 [7].

Additional studies show the re-distribution of claudin-1 between membrane and cytoplasmic locations when the same cell line was treated with STb, with 40% more claudin-1 present within the cytoplasm compared to controls. This loss of membrane claudin-1 was attributed to a dephosphorylation of this TJ protein from the complex (Figure 3b) [24]. Claudin-1 has also been shown to regulate barrier homeostasis through the modulation of Notch-signaling, inducing colonic epithelial proliferation Notch-dependently [25]. The relationship between STb intoxication and claudin-1 Notch-signaling modulation is a new area of research, which could shed light on apparent apoptosis and claudin-1 dysregulation. Apart from claudin-1, claudin-2 has been shown be an employable target in immune-mediated diseases such as inflammatory bowel disease (IBS) [26]. *C. concisus* bacterial invasion has been shown to cause epithelial sodium channel (ENaC) dysfunction as well as claduin-8-dependent barrier dysfunction, both alterations contributing to Na^+^ malabsorption and leak flux diarrhea [27]. The effect STb may have on these crucial barrier-upholding proteins needs further attention, as the literature is limited to claudin-1. All of these combined TJ alterations during an ETEC infection by the STb toxin are involved with disrupting intestinal barrier function, impairing paracellular flux mechanisms and increasing the permeability of the epithelium. These changes all contribute to the observed diarrhea within the host. The degradation of TJ proteins can have downstream effects to the host, leading to other forms of enteric pathology. Long-term effects of TJ disruption and degradation in the host needs to be further investigated.

### 2.2. STb Pore Formation and Internalization by Enterocytes

As mentioned earlier, the luminal surface of the intestine consists of a dynamic barrier of epithelial cells, which impede the transport of materials through paracellular flux mechanisms [18,23]. This epithelial barrier must maintain efficient uptake of nutrients, while also preventing the absorption of unwanted luminal antigens, microorganism and toxins [28]. Bacteria and their subsequent products have derived methods in order to penetrate this barrier such as TJ alteration, known as paracellular translocation. This is when the bacteria or its products traverse between adjacent cells through TJ protein manipulation [29]. TJ proteins normally prevent paracellular passage of bacteria, however, in some cases their toxins, such as STb, have been shown to dislodge crucial TJ proteins, allowing for paracellular passage [24,29]. Alternatively, products may pass through the enterocytes themselves. This is known as transcellular passage, either through endocytosis, internalization or pore formation within the plasma membrane. Studies have shown the presence of bacteria and their toxins within the cytoplasm of enterocytes, measuring translocation in vivo. where the enterocyte monolayers remained unimpaired [29].

There are a number of toxins that undergo oligomerization in order to form pores within the plasma membrane of target cells. This process is crucial for the internalization of the toxin and subsequent intoxication of the cell. These oligomeric channels can either form prior to membrane insertion or once its monomeric peptides are inserted into the membrane [30]. These processes can also occur for smaller peptides of 20–50 amino acid residues, forming at least tetramers depending on the size of the bacterial peptide [31]. Knowing that STb is comprised of an amphipathic (Cys10-Lys22) and hydrophobic (Gly38-Ala44) α-helix, it has been suggested that it can potentially insert itself into the plasma membrane and form pores through oligomerization [32]. The first STb oligomerization characteristic was reported by Labrie et al. (2000) [31]. STb peptides formed hexamers and heptamers independent of temperature and sulfatide receptor binding. STb structural integrity was essential for its oligomerization, as reduced conditions in the presence of β-mercaptoethanol and detergent prevented oligomerization. Site-directed mutations lowering STb hydrophobicity within its hydrophobic α-helix portion rendered the toxin unable to form oligomers. This suggests that the C-terminal hydrophobic α-helix portion of the STb toxin is associated with the domain responsible for STb-STb inter-binding [31]. As mentioned, most pore forming toxins must undergo an oligomerization step. The aerolysin toxin of *Aeromonas hydrophila* forms an oligomeric channel prior to insertion within the plasma membrane [33]. Alternatively, the *Staphylococcus aureus* α-hemolysin toxin undergoes oligomerization once its monomeric formations are inserted into the plasma membrane [34]. STb oligomerization and pore formation seems to be an integral mechanism for STb intoxication as a whole. Studying the underlying mechanism that allows STb to become inserted within the membrane must be further reviewed, as mitigating this step may be essential to combating cellular permeabilization.

A later study demonstrated the internalization of STb utilizing an anti-toxin gold labeling assay and electron microscopy [35]. The fusion STb protein was internalized within rat enterocytes, while a mutant STb with reduced hydrophobicity did not become internalized. The gold particles did not aim at any particular subcellular compartments within the enterocytes, showing a random distribution of STb within the cell after internalization [35]. Alternatively, a later study revealed STb clusters within an NIH-3T3 murine cell line that matched with mitochondria labelling. Mitochondria hyperpolarization, an initial event of intoxication, was observed after STb cell treatment, increasing dose dependently, also permeabilizing the plasma membrane [36]. Moreover, electrophysiological studies utilizing artificial planar lipid bilayers demonstrated the ability of STb to form voltage-dependent ionic pores independent of sulfatide receptor binding [37]. The ability of STb to form pores has not been further investigated since these early studies. This step of oligomerization and pore formation permits cell intoxication, and, as such, needs to be further elucidated. Various therapies may need to be generated in order to reduce STb pore formation, either by enhancing the enterocyte plasma membrane or by mitigating STb oligomerization before pore formation. Knowing whether or not the oligomerization of STb occurs prior to insertion or once the monomer peptides are inserted would also be an asset to future related research. 

### 2.3. STb Triggers Enterocyte Programmed Cell Death

Intestinal cellular proliferation must be counterbalanced by programmed cell death in order to maintain a stable growth and density of enterocytes [38]. Apoptosis occurs involuntarily, maintaining a balance between surviving cells and newly proliferated ones. This processes is characterized by morphological and biochemical alterations, such as cell shrinking, DNA fragmentation and membrane blebbing [38]. The death of these cells has little effect on intestinal barrier function, as a balance of live and dead cells are constantly maintained [39]. Key regulators for apoptosis are a family of cysteine proteases known as caspases, acting as common death effector molecules [40]. 

Apoptosis is activated either through extrinsic or intrinsic pathways. Extrinsic apoptosis is activated by the binding of extracellular death ligands to their respective cell death receptors. The binding of these ligands to their receptors triggers a certain death domain binding. A downstream target then associates with procaspase-8, leading to its activation into caspase-8, the extrinsic apoptosis executioner caspase. This then cleaves procaspase-3 into active caspase-3 by activating a downstream cascade leading to cell death [29,30,31,35,36]. However, intrinsic apoptosis activates independently of ligand and transmembrane receptor binding. BH3-only apoptotic activating proteins and downstream targets activate mitochondrial outer membrane permeabilization (MOMP), releasing mitochondrial cytochrome C into the cytoplasm. This heme protein and downstream apoptosis protease activating factor-1 (APAF-1) then oligomerize into an apoptosome through procaspase-9 interaction. Procaspase-9 is then cleaved into caspase-9, leading to caspase-3 initiation and endonuclease DNA degradation and cell death [38,39,40,41,42,43].

The intestinal mucosa is the main harbor and initial interaction site for pathogenic microorganisms [40]. Intrinsic and extrinsic pathways can both be triggered by pathogens themselves, such as ETEC, or by their derived products, such as the STb toxin. It was found that caspase-9, the mitochondrion-mediated intrinsic initiator, and its effector caspase-3 were both triggered by STb within a mouse and human cell line. Caspase-8, the extrinsic initiator, was not activated however, suggesting that STb activates intrinsic, caspase-dependent apoptosis [9]. The death of these cells may be responsible for the loss of mucosal surface area and luminal fluid accumulation, suggesting that STb-mediated apoptosis and subsequent diarrhea are linked. 

With regards to ETEC infection as a whole, Xia et al. (2018) demonstrated how ETEC infection in vivo actually inhibited the activation of caspase-9 mitochondrial-mediated, intrinsic apoptosis. ETEC was found to utilize and activate Caspase-8 extrinsic apoptosis instead. This suggests that, during an ETEC infection, intrinsic apoptosis becomes inhibited, while extrinsic apoptosis becomes activated and utilized to cause cell death [44]. The reasoning for this discrepancy is still unknown. A number of factors can be attributed to this, such as the various other toxins secreted during ETEC infection, as well as the effect the bacteria itself, may have on intestinal epithelial cells. Additionally, internal signals initiating intrinsic apoptosis can be attributed to the internalization of STb specifically. Cell death by STb intoxication can be a great threat to host intestinal health. The death of these cells and their long-term effects must be examined. 

### 2.4. STb Causes Deterioration of the Intestinal Absorptive Mucosa

Absorption and secretion is a simultaneous physiological process within the mammalian intestinal tract. The recirculation of water and ions from the intestinal lumen into the enterocyte can occur either trans-or-paracellularly. Water influx can occur through diffusion directed by osmotic gradients during ion exchange, or by the hydration of solutes which then enter absorptive cells through various transporters [45,46]. This will ensure isotonicity of luminal contents prior to absorption [47]. However, if water efflux surpasses water influx, a net secretory state occurs within the intestine, leading to malabsorption and most often diarrhea [45]. The majority of nutrient transport occurs within the small intestine, while the large intestine is generally responsible for water and ion exchange [48]. The small intestinal surface area is dramatically enlarged by miniature projections of villi and microvilli. They are covered with absorptive columnar epithelial cells at the tip, while crypt cells are most often secretory. Nearly all ingested nutrients are absorbed into the blood stream through this exceptionally polarized layer of epithelial cells, which forms the intestinal mucosa [45,46,48]. 

The deterioration of the intestinal mucosa, such as the loss of absorptive cells and decreased villus height, can be attributed to alterations in the flux of water and ions within the lumen. The first report of cellular alterations caused by STb was conducted by Whipp et al. (1985) [49]. They demonstrated alterations of villous length by STb within pig intestinal loops, however having no effect on crypt depth [49]. STb significantly lowered the rate of net fluid and ion absorption. The loss of the intestinal villi epithelium through virus manipulation experiments (transmissible gastroenteritis virus) substantially decreased the response to STb. This suggests that differentiated villous epithelium must be present in order for a maximal STb response to occur [49]. Following this study, Whipp et al. (1986) [50] showed that STb exposure to pig intestinal loops brought about microscopic alterations of the intestinal mucosa and its structural integrity. This included decreased villous height (11–19% shrinkage, including loss and atrophy of villous absorptive cells) and an increased amount of sloughed epithelial cells within the lumen. Additionally, a shift in the morphology of the cells from columnar to cuboidal epithelium upon the affected villi was observed. This was accompanied by an increased incidence of disrupted epithelium upon the affected villous, accordant with a compromised absorptive capability [50]. 

In a follow-up study by Whipp et al. (1987) (Figure 4) [51], STb was attributed to morphological alterations and lesions correlated to a loss and atrophy of villous absorptive cells. The crypt surface area showed no significant difference when treated with STb. The intestinal mucosal surface area was 20% less when treated with STb compared to controls. STb exposure also lowered sucrase activity by 15%, with this observation being correlated with villous atrophy [51]. These combined observations by Whipp and colleagues make it apparent that STb induces lesions within the intestinal epithelium, suggesting a loss of absorptive mucosal cells. The deterioration of the abortive mucosa can have many direct and indirect effects on gut health. STb alteration of the mucosa can debilitate epithelial absorption of luminal nutrients and can induce a net secretion, contributing to malabsorption and diarrhea. 

### 2.5. Fate of Enteroendocrine Cells and Gut Hormone Regulation

There is a great deal of knowledge of the effects of STb on the intestinal epithelium. This epithelium consists of absorptive enterocytes, making up the mucosal barrier. Dispersed among these mucosal cells, specialized enteroendocrine epithelial cells can also be found. These cells represent approximately 1% of the entire gastrointestinal epithelium [52]; however, when counted throughout the GI tract, they make up the largest endocrine organ in the body. These cells produce and secrete a wide variety of regulatory hormones and signaling molecules which function to complete different tasks [53]. They regulate functions such as appetite, contraction and motility of various organs, intestinal growth and barrier enhancement, as well as glucose homeostasis. Within the small and large intestines, enteroendocrine L cells make up the majority of endocrine cells within the gastrointestinal tract [52,53]. These cells secrete glucagon-like peptide 1 (GLP-1), a 30 amino acid peptide hormone produced through differential processing of proglucagon by prohormone convertase 1/3 (PC1/3) [54]. The secretion of GLP-1 is stimulated by the ingestion of glucose and various other nutrients. GLP-1 enhances glucose dependent insulin secretion from β-pancreatic cells [54,55]. With the additional effect of inhibiting glucagon secretion from the pancreas, GLP-1 functions to lower blood glucose levels after a meal and inhibits the conversion of glycogen reserves into glucose [56]. GLP-1 is an anorexigenic hormone, promoting satiety and slowing gastric emptying, allowing for nutrients to be fully absorbed within the proximal gut [57]. Interestingly, GLP-1 receptor agonists are used for the treatment of Type 2 diabetes mellitus (T2DM) and obesity, due to its ability to reduce blood glucose levels and lower body weight [54]. GLP-1 has also been noted to improve inflammatory macrophage-derived insulin resistance through the inhibition of the NF-κB pathway and the secretion of inflammatory cytokines in macrophages [58].

Co-secreted with GLP-1 is Glucagon-like peptide 2 (GLP-2), a peptide hormone of 33 amino acid residues, also processed from proglucagon [59]. GLP-2 is a potent intestinotrophic growth factor, inducing crypt cell proliferation and inhibiting apoptosis of intestinal epithelial cells. This results in increased villous height, as well as greater absorptive mucosal surface area within the gut [59]. GLP-2 has been observed to significantly reduce intestinal fluxes of Na^+^, bacterial epithelial penetration, and inflammatory colonic cells [60]. With these combined effects under stress conditions, GLP-2 can significantly enhance intestinal barrier function and reduce the penetration of unwanted luminal antigens [61]. The enhancement of the intestinal barrier by GLP-2 can be attributed to an increased expression of TJ proteins. Yu et al. (2014) revealed how GLP-2 significantly increased the mRNA and protein expressions of ZO-1, occludin and claudin-1. A mitogen-activated protein kinase (MAPK) pathway inhibitor in conjunction with GLP-2 mitigated the same mRNA and protein expressions. This suggests that GLP-2 improves the expression of crucial TJ proteins potentially through the MAPK signaling pathway [62]. GLP-2 has also been noted to act as a protective factor against the deregulation of glucose metabolism which occurs in obese conditions [59]. Importantly, GLP-2 is approved for use in intestinal malabsorption diseases, including Crohn’s disease and short bowel syndrome (SBS) [63]. Teduglutide is a potent analogue of GLP-2, having a half-life of around 25 times that of natural GLP-2 [64]. Newly emerging brands such as Gattex employ this analogue to treat SBS symptoms like nausea and severe diarrhea [63], similar to the symptoms caused by the STb toxin.

As described throughout, the STb toxin has numerous harmful effects on intestinal cells and the absorptive mucosa, impairing many gut functions. Intestinal endocrine cells and intestinal epithelial cells share similar morphologies, such as the STb receptor, sulfatide, dispersed on the cell surface, as well as downstream targets. Once STb is bound, there is no known mechanisms, which would mitigate STb from becoming internalized, inducing apoptosis or fluid secretion. Enteroendocrine L cells would be at most risk during an ETEC infection, as the bacteria tends to harbor within the distal gut. The effect of STb on the secretion of GLP-1 and GLP-2, as well as the physiological alterations to these endocrine cells, must be elucidated. These cells are vulnerable to intoxication, so one might question the fate of these cells, as well as the fate of gut hormone regulation as a whole during an ETEC infection. Research has yet to shine its light on this area of study, as intestinal hormone deregulation could have underlying effects on top of the observed effects of an ETEC infection.

## 3. Conclusions

From the data discussed thus far, it is evident that new areas and routes for the STb toxin are emerging and future research must occur to uncover the exact mode of action of STb and its effects. Purification techniques, cell culturing and many other methodologies have improved within the past decades, allowing for a clearer and deeper understanding of new STb-related actions. There are several STb-induced effects which can be attributed to alternative STb-related mechanisms. For example, fluid and ion loss can be attributed to a number of mechanisms other than ionic transporter activation. One example of this is the paracellular loss of fluids and ions through tight junction manipulation. Another is the transcellular loss of fluids and ions through cellular permeabilization or plasma membrane pore formation. The significance of each effect relative to its proposed mechanism needs to be further determined, in order to understand their contribution within the process. The ability of STb to conduct such actions in a non-specific, almost mechanistic manner is dangerous, as almost all gut cells from a wide variety of species are vulnerable to STb intoxication. However, the ability of STb to form pores and manipulate tight junctions non-specifically can be harnessed for example in drug delivery. Gut hormone dysregulation from bacterial toxins is a novel research area which needs greater attention. GLP-2 possesses properties opposite to that of the STb toxin. Hormone therapy following ETEC infection may become a useful addition to the existing ETEC related therapies. All of these different research areas must be collectively studied, in order to finally take control of global ETEC infection. 

## Figures and Tables

**Figure 1 toxins-12-00760-f001:**
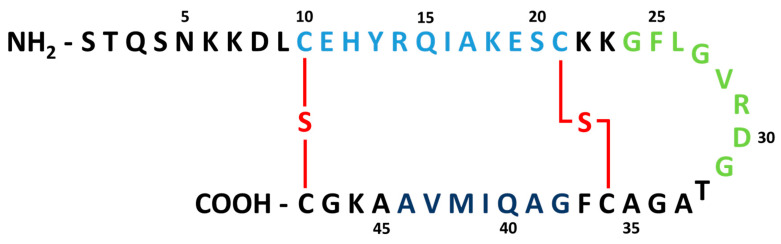
Amino acid skeleton of the mature heat-stable enterotoxin b (STb) toxin. The toxin is comprised of 48 amino acid residues, starting at the N-terminal. Disulfide bridges are shown in red. The integrity of the disulfide bond at Cys 21 and Cys 36 is necessary for the toxigenic region to be active. Light blue residues 10–21: amphipathic alpha-helix portion; residues 18–30: receptor binding/toxigenic region; green residues 24–31: tight junction manipulation region; residues 37–42: oligomerization region; dark blue residues 38–44: hydrophobic alpha-helix region [12,13].

**Figure 2 toxins-12-00760-f002:**
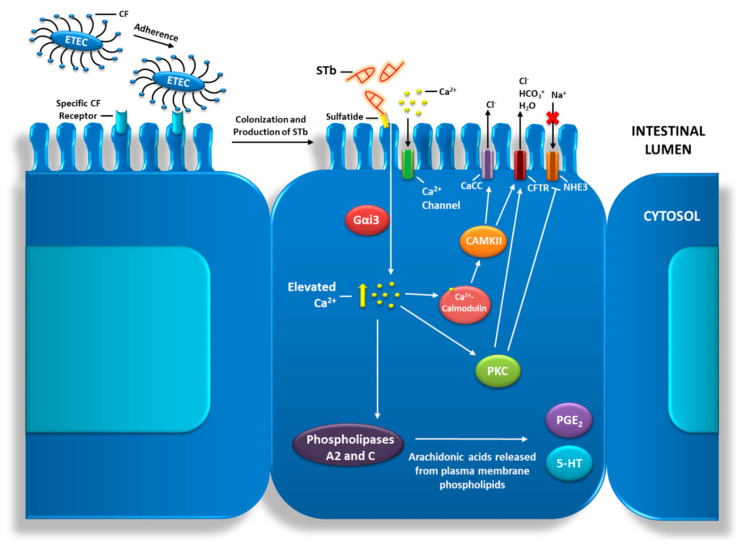
Pathogenesis of Enterotoxigenic *Escherichia coli* (ETEC) and mechanism of action of the STb toxin. ETEC adherence and signaling pathway of the STb toxin leading to water and electrolyte secretion. CF: colonization factor; Gαi3: pertussis toxin-sensitive GTP-binding regulatory protein; CAMKII: calmodulin-dependent protein kinase II; CaCC: calcium-activated chloride channel; PKC: Protein kinase C; CFTR: cystic fibrosis transmembrane conductance regulator; NHE3: Na^+^/H^+^-exchanger 3; PGE_2_: prostaglandin E_2_; 5-HT: 5-hydroxytryptamine.

**Figure 3 toxins-12-00760-f003:**
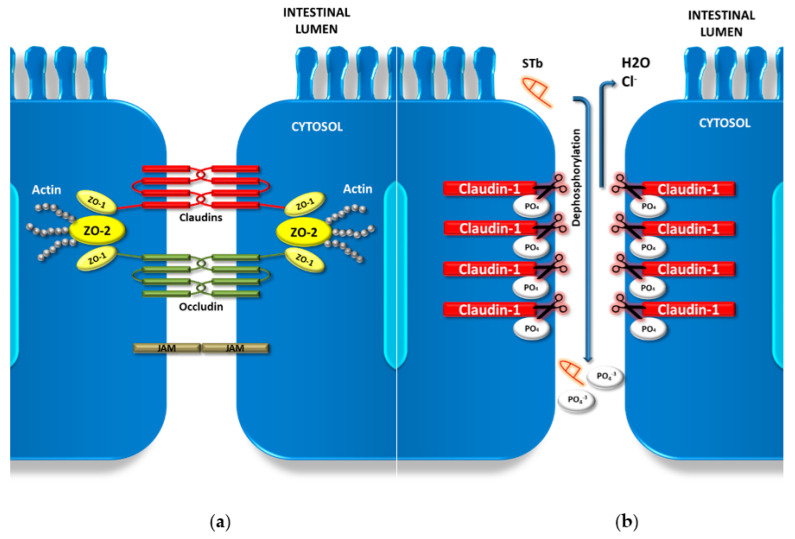
Enterocyte tight junction (TJ) complex and dephosphorylation of claudin-1. (**a**) Schematic of the TJ complex between adjoining cells showing junctional adhesion molecules JAM proteins and the claudin and occludin transmembrane complexes supported by zonula occludens protein (ZO) complexes and the cytoskeleton. ZO-1/2: zonula occludens protein 1/2; JAM: junctional adhesion molecules. (**b**) Mechanism of action of STb-induced claudin-1 dephosphorylation. Claudin-1 moves from the transmembrane position into the cytoplasm following STb dephosphorylation [12].

**Figure 4 toxins-12-00760-f004:**
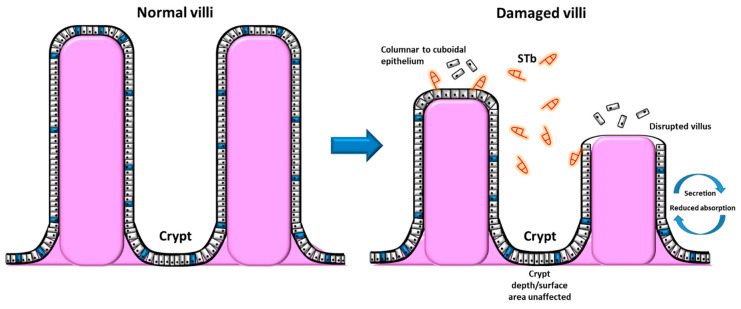
Normal intestinal villi compared to STb-damaged villi showing morphological alterations and villus disruption, however no alteration in crypt depth or surface area [12,51].

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
