# Peer review of "Impact of the Escherichia coli Heat-Stable Enterotoxin b (STb) on Gut Health and Function"

_toxins, 2020, doi:10.3390/toxins12120760_

Round 1
Reviewer 1 Report
Review Report
Impact of the Escherichia coli Heat-Stable Enterotoxin 2 b (STb) on Gut Health and Function
Summary
In this manuscript the authors review the mechanisms by which Heat-Stable Enterotoxin 2 b (STb) produced by enterotoxigenic Escherichia coli (ETEC) affects the integrity of the intestinal epithelium and suggest an exciting new direction for future research investigating the effect of STb on gut hormone regulation.
Specific comments
The review is very well written and structured and it needs only minor corrections listed below.
Line 31: please replace ‘heat-liable’ with ‘heat-labile’
Line 40: What does ‘negative’ refer to? Please rephrase
Line 60: please include ‘(CaCC)’ after calcium-activated chloride channel
Line 62: ‘HCO3+’ is ‘HCO3-‘ in Figure 2. Please amend accordingly.
Line 65: would be useful to include release of arachidonic acids in Figure 2
Line 66: please include abbreviations after prostaglandin E2 and 5-hydrocytryptamine
Line 94: please delete ‘are’ after ‘junctions’
Line 110: please insert numbered reference after Ngendahayo & Dubreuil (2013). Please also do the same for lines 153, 230 and 235
Lines 116-118: These two sentences should be merged
Line 149: please replace ‘on’ with ‘of’
Line 151: please replace ‘this’ with ‘these’
Line 153: ‘first STb oligomerisation characteristic’ would sound better rephrased
Line 160: Please insert reference at the end of this sentence
Lines 187, 188, 194: please replace ‘procasepase’ with ‘procaspase’
Line 267: please replace ‘enteroedocrine’ with ‘enteroendocrine’
Line 302: please delete ‘a’
Figures
Figure 1: Some residues are colour-coded but the colour is not referenced in the Figure legend. For example, including ‘(in blue)’ after ‘residues 10-21’; ‘(in green)’ after ‘residues 24-31; and ‘dark blue’ after ‘residues 38-44’ would help the reader locate these residues easier. Note, the colour of the hydrophobic alpha-helix region is now listed in dark blue and should also be changed in figure.
Figure 2: Text inside cell should be bigger and white against blue would show better.
Reviewer 1 Report
Impact of the Escherichia coli Heat-Stable Enterotoxin 2 b (STb) on Gut Health and Function
Summary
In this manuscript the authors review the mechanisms by which Heat-Stable Enterotoxin 2 b (STb) produced by enterotoxigenic Escherichia coli (ETEC) affects the integrity of the intestinal epithelium and suggest an exciting new direction for future research investigating the effect of STb on gut hormone regulation.
Specific comments
The review is very well written and structured and it needs only minor corrections listed below.
Line 31: please replace ‘heat-liable’ with ‘heat-labile’
- Done, replaced liable with labile
Line 40: What does ‘negative’ refer to? Please rephrase
- Sentence updated to clarify term
Line 60: please include ‘(CaCC)’ after calcium-activated chloride channel
- CaCC abbreviation added
Line 62: ‘HCO3+’ is ‘HCO3-‘ in Figure 2. Please amend accordingly.
- Figure amended accordingly, corrected to HCO3+
Line 65: would be useful to include release of arachidonic acids in Figure 2
- Figure amended accordingly, included written step within figure
Line 66: please include abbreviations after prostaglandin E2 and 5-hydrocytryptamine
- Abbreviations added for both (PGE2 and 5-HT)
Line 94: please delete ‘are’ after ‘junctions’
- Done, ‘are’ deleted after ‘junctions’
Line 110: please insert numbered reference after Ngendahayo & Dubreuil (2013). Please also do the same for lines 153, 230 and 235
- Done, added numbered references after in-text references for every line number mentioned
Lines 116-118: These two sentences should be merged
- Done, lines merged to flow better
Line 149: please replace ‘on’ with ‘of’
- Done, replaced ‘on’ with ‘of’
Line 151: please replace ‘this’ with ‘these’
- Done, replaced ‘this’ with ‘these
Line 153: ‘first STb oligomerisation characteristic’ would sound better rephrased
- Done, sentence rephrased
Line 160: Please insert reference at the end of this sentence
- Done, appropriate reference inserted at end of sentence
Lines 187, 188, 194: please replace ‘procasepase’ with ‘procaspase’
- Done, corrected all terms to ‘procaspase’
Line 267: please replace ‘enteroedocrine’ with ‘enteroendocrine’
- Done, corrected spelling
Line 302: please delete ‘a’
- Done, deleted ‘a’
Figures
Figure 1: Some residues are colour-coded but the colour is not referenced in the Figure legend. For example, including ‘(in blue)’ after ‘residues 10-21’; ‘(in green)’ after ‘residues 24-31; and ‘dark blue’ after ‘residues 38-44’ would help the reader locate these residues easier. Note, the colour of the hydrophobic alpha-helix region is now listed in dark blue and should also be changed in figure.
- Figure fixed, residues 38-44 switched to dark blue instead of light blue, all coloured residues stated within legend before stating numbering
Figure 2: Text inside cell should be bigger and white against blue would show better.
- Fire fixed, all text within cell switched to white, all test size increasing inside and outside of cell, arrow widths also increased
Reviewer 2 Report
The heat-stable enterotoxin b (STb) is a toxic protein produced and secreted by enterotoxigenic Escherichia coli. Authors of this paper describe and summarize the mechanisms of STb action, focusing on processes that alter the gastrointestinal tract. In my opinion, these mechanisms are described very briefly and the information about the mode of action of STb are presented too broad. In the abstract, the authors point out that their attention will be focused on the pore-forming activity and internalization processes. Unfortunately, these threads are described very concisely- these mechanisms are not compared with other proteins and are not discussed in details. I think that this review paper requires significant changes before reconsideration and acceptance for publication in the Toxin journal.
Reviewer 2 Report
Comments and Suggestions for Authors
The heat-stable enterotoxin b (STb) is a toxic protein produced and secreted by enterotoxigenic Escherichia coli. Authors of this paper describe and summarize the mechanisms of STb action, focusing on processes that alter the gastrointestinal tract.
In my opinion, these mechanisms are described very briefly and the information about the mode of action of STb are presented too broad. In the abstract, the authors point out that their attention will be focused on the pore-forming activity and internalization processes. Unfortunately, these threads are described very concisely- these mechanisms are not compared with other proteins and are not discussed in details.
- We acknowledge that additional depth on mechanistic pathways is available for each of the areas presented. We appreciate that this review does not delve into details on comparisons with other proteins. We have added a sentence to the introduction to guide readers to appropriate reviews for this information (Line 33).
Reviewer 3 Report
In this review, the authors have provided a concise and well-focused overview of the impact of the E coli heat stable enterotoxin b in intestinal barrier function, pore formation, intestinal absorption, apoptosis and hormone generation. Particularly the discussion about interactions between GLP-2 and Stb is novel.
I only have some minor recommendations for improvement.
- It could be nice to shortly discuss the interaction between the heat labile and heat stable toxins, since they are introduced at the end of the first paragraph in the introduction. On line 31 labile is misspelled.
- In figure 2, it could be appropriate to also include the Na+-K+-pump at the basolateral side of the cells, which is also involved in H2O transport through Na+ secretion.
- In line 135, the sentence should be rephrased. TER is a method to measure barrier function, but is not a mechanism to impede material transport, like e.g. paracellular flux.
- Procaspase 9 is misspelled in line 194.
- In Fig. 3b, it would be better to specify Claudin-1, since other claudins exist which have different roles. As such, claudin-2 was recently shown to promote experimental colitis (Raju et al., J Clin Invest 2020). It could also be interesting to discuss other Claudins which have been shown to regulate intestinal barrier function, like Claudin-8. The effect of a reduction in Claudin expression in TJs on back leakage of cations and anions into the intestinal lumen could also be discussed. This was recently shown to be the case for Claudin-8 (Nattramilarasu et al., IJMS 2020).
- The role of Claudin 1 in intestinal epithelial homeostasis through the modulation of Notch signaling was not discussed (Pope et al., Gut 2014) and could provide a nice transition between the barrier function and proliferation/apoptosis sections.
Author Response
Reviewer 3
Comments and Suggestions for Authors
In this review, the authors have provided a concise and well-focused overview of the impact of the E coli heat stable enterotoxin b in intestinal barrier function, pore formation, intestinal absorption, apoptosis and hormone generation. Particularly the discussion about interactions between GLP-2 and Stb is novel.
I only have some minor recommendations for improvement.
- It could be nice to shortly discuss the interaction between the heat labile and heat stable toxins, since they are introduced at the end of the first paragraph in the introduction. On line 31 labile is misspelled.
- We agree that this review does not delve into details on comparisons with other enterotoxins. We have added a sentence to the introduction to guide readers to appropriate reviews for this information (Line 33). Labile spelling corrected as well.
- In figure 2, it could be appropriate to also include the Na+-K+-pump at the basolateral side of the cells, which is also involved in H2O transport through Na+ secretion.
- We appreciate this suggestion, however we felt that since the opening of the K+ channel occurs primarily during STa intoxication instead of STb intoxication we chose not to include it. This was based on our review of this study. “Mechanism of action of Escherichia coli heat stable enterotoxin in a human colonic cell line” DOI: 1172/JCI113626
- In line 135, the sentence should be rephrased. TER is a method to measure barrier function, but is not a mechanism to impede material transport, like e.g. paracellular flux.
- Done, line 135 fixed, removed incorrect term from sentence
- Procaspase 9 is misspelled in line 194.
- Done, all procaspase misspellings corrected
- In Fig. 3b, it would be better to specify Claudin-1, since other claudins exist which have different roles. As such, claudin-2 was recently shown to promote experimental colitis (Raju et al., J Clin Invest 2020). It could also be interesting to discuss other Claudins which have been shown to regulate intestinal barrier function, like Claudin-8. The effect of a reduction in Claudin expression in TJs on back leakage of cations and anions into the intestinal lumen could also be discussed. This was recently shown to be the case for Claudin-8 (Nattramilarasu et al., IJMS 2020).
- Figure fixed, changed to Claudin-1 specifically within figure. Lines 127-131 added and discussed Claudin-2 as well as Claudin-8, cited specific papers given.
- The role of Claudin 1 in intestinal epithelial homeostasis through the modulation of Notch signaling was not discussed (Pope et al., Gut 2014) and could provide a nice transition between the barrier function and proliferation/apoptosis sections.
- Lines 123-127, discussed the role of Claudin-1 Notch-signalling modulation and what it may mean in terms of STb intoxication and STb Claudin-1 dysregulation as well as apoptosis in general, referenced specific article given.
Round 2
Reviewer 2 Report
A sentence added by the authors to the introduction section is very helpful for readers. Other corrections added to this manuscript make this review more clear and understandable. In my opinion, the part concerning the mechanism of pore formation of STb is still poorly described. Addition of a little information about the pore structure or possible mechanism of pore formation of STb will be beneficial to the overall value of this review. Could the authors compare the mechanism of action of STb with other pore-forming toxins? Short discussion about possible mechanisms of pore formation of STb will be a big advantage of this paper.
Author Response
Thank you for your re-examination of our review article. We have added additional details on pore formation (lines 163-166) as well as comparisons in oligomerization with other toxins(172-179).